# Agent Planning with World Knowledge Model

**Shuofei Qiao**♠,*, **Runnan Fang**♠,*, **Ningyu Zhang**♠,†, **Yuqi Zhu**♠, **Xiang Chen**♠,
**Shumin Deng**♣, **Yong Jiang**◇, **Pengjun Xie**◇, **Fei Huang**◇, **Huajun Chen**♠♡†
♠Zhejiang University  ♣National University of Singapore, NUS-NCS Joint Lab  ◇Alibaba Group
♡Zhejiang Key Laboratory of Big Data Intelligent Computing
{shuofei,zhangningyu}@zju.edu.cn

## Abstract

Recent endeavors towards directly using large language models (LLMs) as agent models to execute interactive planning tasks have shown commendable results. Despite their achievements, however, they still struggle with brainless trial-and-error in global planning and generating hallucinatory actions in local planning due to their poor understanding of the "real" physical world. Imitating humans' *mental* world knowledge model which provides global prior knowledge before the task and maintains local dynamic knowledge during the task, in this paper, we introduce *parametric* **W**orld **K**nowledge **M**odel (**WKM**) to facilitate agent planning. Concretely, we steer the agent model to self-synthesize knowledge from both expert and sampled trajectories. Then we develop WKM, providing prior *task knowledge* to guide the global planning and dynamic *state knowledge* to assist the local planning. Experimental results on three complex real-world simulated datasets with three state-of-the-art open-source LLMs, Mistral-7B, Gemma-7B, and Llama-3-8B, demonstrate that our method can achieve superior performance compared to various strong baselines. Other interesting findings include: 1) our instance-level task knowledge can generalize better to unseen tasks, 2) weak WKM can guide strong agent model planning, and 3) unified WKM training has promising potential for further development[3].

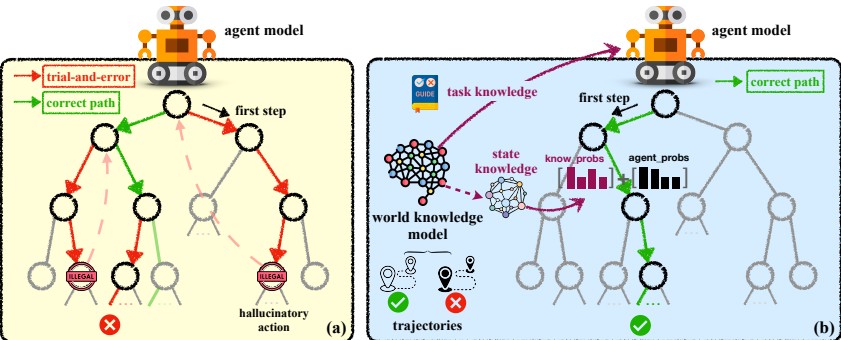

Figure 1: Traditional agent planning vs. Agent planning with world knowledge model.

## 1 Introduction

The remarkable advances in Large Language Models (LLMs) have witnessed a rapid development of various natural language processing tasks [25, 16, 28, 47, 60, 33]. Recently, multiple attempts that

---

\*     Equal Contribution.
†     Corresponding Author.
[3]The code is available at https://github.com/zjunlp/WKM.

38th Conference on Neural Information Processing Systems (NeurIPS 2024).

directly exploit LLMs as agent models to address physical world planning tasks have demonstrated promising achievements [54, 57, 56, 34, 38, 64, 44]. However, as most state-of-the-art LLMs are autoregressive models trained with next-token prediction, they lack the ability to essentially understand the real world, leading to generating hallucinatory actions and performing brainless trial-and-error in the environment as shown in Figure 1(a).

In contrast to LLMs, humans possess a mental knowledge model about the physical world [1, 18, 17, 30]. When facing a specific task, they will first briefly rehearse the entire process in mind using their rich prior knowledge before performing mindless actions. We call this kind of knowledge global *task knowledge* (a.k.a. environment/task commonsense). In addition, during the task procedure, the mental world knowledge model will constantly maintain a kind of local *state knowledge*, representing humans' cognition of the current world state. For example, imagine you are in a room and your task is to `put a clean egg in microwave`. The *task knowledge* may refer to `The egg is most likely in the fridge ... The workflows are: 1) locate and take the egg; 2) clean the egg using sinkbasin ...` The *state knowledge* possibly refers to `My task is to ... I have found and taked the egg ... Next I should ...` The absence of world knowledge can lead to blind trial-and-error in the early planning stages when environmental information is limited. Conversely, in later stages when information is redundant, it can easily result in a confused cognition of the current world state and generate hallucinatory actions.

The process by which humans handle planning tasks reminds us to develop a parametric **W**orld **K**nowledge **M**odel (**WKM**) to facilitate agent planning. As humans typically acquire knowledge from expertise and practical experience, we build WKM based on knowledge learned from both expert and explored trajectories. Specifically, we first steer the agent model to synthesize task knowledge from the comparison between expert and sampled trajectories. Then we prompt it to summarize state knowledge for each planning step from expert trajectories and combine the previous and next actions to build a state knowledge base. Lastly, we integrate the generated knowledge into expert trajectories and train a WKM. The agent model needs to be retrained to adapt to the task knowledge. Note our agent and knowledge model are both trained with LoRA [12] sharing the same backbone.

During the planning phase, we use the WKM to provide global prior task knowledge and maintain local dynamic state knowledge for the agent model as shown in Figure 1(b). The task knowledge will be concatenated in natural language form following the specific task to guide the agent model's trial-and-error. At each planning step, to prevent the occurrence of hallucinatory actions, we utilize the generated state knowledge as the query to conduct $k$NN retrieval from the pre-built state knowledge base. We then use the constraints from the previous action, the probabilities of the retrieved next actions, and the probabilities from the agent model to make a weighted prediction for the next action.

We evaluate our method on three real-world simulated planning tasks: ALFWorld [41], WebShop [53], and ScienceWorld [50] with three state-of-the-art open-source LLMs: Mistral-7B [16], Gemma-7B [24], and Llama-3-8B [25]. Empirical results demonstrate that our method achieves superior performance compared to various strong baselines on both seen and unseen tasks. Moreover, further analytical results show that 1) our WKM can effectively reduce blind trial-and-error and hallucinatory actions, 2) our model-generated instance-level knowledge can generalize better to unseen tasks, 3) weak-guide-strong is feasible, 4) multi-task unified WKM possesses strong potential, and 5) explicit state knowledge will hurt the performance of agent planning.

## 2 Preliminaries

We mainly focus on interactive tasks with partial observations from environments. Following the task formulation in [44], the problem can be viewed as a Partially Observable Markov Decision Process (POMDP): $(\mathcal{U}, \mathcal{S}, \mathcal{A}, \mathcal{O}, \mathcal{T})$. The instruction space $\mathcal{U}$ defines the task and its corresponding regulations. $\mathcal{S}$ is the state space, $\mathcal{O}$ is the observation space, and $\mathcal{A}$ is the action space. $\mathcal{T} : \mathcal{S} \times \mathcal{A} \to \mathcal{S}$ defines the transition function, which we assume to be given by the environments. It is noticed that $\mathcal{U}, \mathcal{A}$, and $\mathcal{O}$ are subspaces of the natural language space in the language agent scenarios.

Based on the above, the historical trajectory $h_t$ that consists of a list of actions and observations at time $t$ can be represented as:
$$h_t = (u, a_0, o_0, a_1, o_1, \ldots, a_t, o_t), \tag{1}$$
where $u \in \mathcal{U}$ is the task instruction and $a \in \mathcal{A}$, $o \in \mathcal{O}$ are the action and the observation. Given a task, the language agent with parameter $\theta$ serves as the policy model $\pi_\theta$ responsible for generating

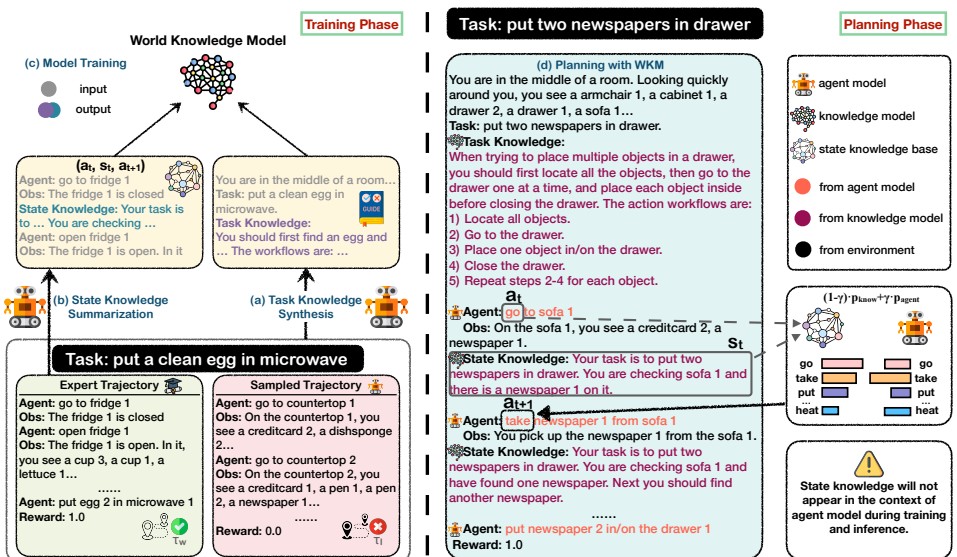

Figure 2: **Overview of our WKM**. We train a world knowledge model on the knowledge synthesized by the agent model itself from both expert and explored trajectories, providing prior task knowledge to guide global planning and dynamic state knowledge to assist local planning.

the action $a_{t+1}$ based on $h_t$ at each time step $t + 1$:

$$a_{t+1} \sim \pi_\theta(\cdot | h_t). \tag{2}$$

Specifically, $a_0 \sim \pi_\theta(\cdot | u)$ is generated according to the task instruction $u$. The whole trajectory $\tau$ concludes when the task is completed or exceeds the maximum time steps. Then the production of the entire trajectory with time length $n$ can be modeled as:

$$\pi_\theta(\tau | u) = \prod_{t=0}^{n} \pi_\theta(a_{t+1} | h_t) \pi_\theta(a_0 | u). \tag{3}$$

Ultimately, the final reward $r(u, \tau) \in [0, 1]$ representing the task completion rate is calculated. Note that we follow a REACT-style [54] trajectory that includes rationales before each action. We use $a$ to represent the action with rationales for convenience.

**World Knowledge Model.** *World knowledge model* serves as humans' mental cognition of the physical environment, more intricate than the *word knowledge model* which LLM-powered agent models are trained to be [61, 10, 52, 13]. Our "world" here refers to the simulated environment of the task. Based on the static environment of the task and the dynamic changes during interaction with the agent, we define world knowledge as a combination of prior global knowledge and dynamic local knowledge, corresponding to the blind trial-and-error problem in global planning and the hallucinatory action issue in local planning in traditional agent models, respectively. To attain precise and efficient agent planning, we develop a *parametric* WKM to simulate the *mental* WKM of humans.

## 3 Method

As shown in Figure 2, we steer the agent model to self-synthesize the *task knowledge* from the comparison of expert and sampled trajectories (§3.1). Then we prompt the agent model to self-summarize the *state knowledge* based on historical behavior and construct a state knowledge base (§3.2). The generated knowledge will be integrated into the expert trajectories for training the WKM. After the training process (§3.3), we augment the agent model with the world knowledge model to achieve effective and accurate planning (§3.4).

### 3.1 Task Knowledge Synthesis

The *task knowledge* serves as the prior knowledge to guide the agent model's global planning and prevent it from dropping into blind trial-and-error.

**Experienced Agent Exploration.** We primarily acquire task knowledge through the comparison of preference trajectories (*chosen* vs. *rejected*). In order to improve the quality of rejected trajectories and obtain more targeted task knowledge, we employ an experienced agent for exploration. Firstly, we train a vanilla language model with expert trajectories[4] from the training set to obtain an experienced agent. Subsequently, the experienced agent explores the training set tasks again to generate rejected trajectories. Our purpose is to extract superior task knowledge that cannot be acquired solely through supervised fine-tuning on chosen trajectories, thus further effectively boosting the agent's capabilities.

**Self Knowledge Synthesis.** With the expert trajectories as the chosen ones and the trajectories sampled from the experienced agent as the rejected ones, we prompt the agent model itself to synthesize the task knowledge. Supposing $\mathcal{K}$ is the task knowledge space:

$$\kappa \sim \pi_\theta(\cdot|\rho_{\text{TaskKnow}}, u, \tau_w, \tau_l), \tag{4}$$

where $\kappa \in \mathcal{K}$ is the task knowledge, $\rho_{\text{TaskKnow}}$ stands for the prompt to instruct the task knowledge extraction, and $\tau_w$, $\tau_l$ are the chosen and rejected trajectories respectively. Note that given the same task $u$, $\tau_w$ and $\tau_l$ always satisfy $r(u, \tau_w) = 1 \geq r(u, \tau_l)$. Even when $r(u, \tau_w) = r(u, \tau_l)$, we still consider trajectories sampled from the experienced agent as rejected ones. This is because expert trajectories often have shorter step lengths, enabling the agent to learn more knowledge of efficient planning. For detailed prompts of task knowledge synthesis, please refer to Appendix I.1.

## 3.2 State Knowledge Summarization

The *state knowledge* serves as the dynamic knowledge to constrain the agent model's local planning and prevent it from generating hallucinatory actions. We prompt the agent model to self-summarize state knowledge at each planning step based on the expert trajectories to guarantee quality. For detailed prompts of state knowledge summarization, please refer to Appendix I.2. Supposing the prompt used to summarize state knowledge is $\rho_{\text{StateKnow}}$ and the state knowledge $s \in \mathcal{S}$ is a part of the state space $\mathcal{S}$, the generation of state knowledge at time $t$ can be represented as:

$$s_t \sim \pi_\theta(\cdot|\rho_{\text{StateKnow}}, h_t). \tag{5}$$

**State Knowledge Base Construction.** To avoid confusion caused by excessive additional information, instead of explicitly concatenating the state knowledge to the context, we construct a state knowledge base for retrieval (we analyze in §4.3 how explicit state knowledge may affect the performance of agent model). We combine the state knowledge $s_t$ with the previous action $a_t$ and next action $a_{t+1}$ from the expert trajectory to form a action-state-action triplet $(a_t, s_t, a_{t+1})$. After iterating through all expert trajectories, we obtain a State Knowledge Base $\mathcal{B} = \{(s, a_{\text{pre}}, a_{\text{next}})^{(i)}\}_{i=1}^{|\mathcal{B}|}$, where $a_{\text{pre}} = a_t$, $a_{\text{next}} = a_{t+1}$, and $|\mathcal{B}|$ is the size of the state knowledge base.

## 3.3 Model Training

We integrate the generated world knowledge into expert trajectories and train a world knowledge model. The agent model needs to be re-trained to adapt to the incorporation of task knowledge. Note that our agent model and knowledge model are both trained with LoRA **sharing the same backbone**. We list the examples of training data for both the agent model and WKM in Appendix E.

**Agent Model Training.** Given the expert trajectories dataset $\mathcal{D} = \{(u, \kappa, \tau_w)^{(i)}\}_{i=1}^{|\mathcal{D}|}$ with **task knowledge** $\kappa$ generated in §3.1, we train the agent model to follow the task knowledge to generate actions. Under an auto-regressive manner, the loss of the agent model can be formulated as:

$$\mathcal{L}_{\text{agent}}(\pi_\theta) = -\mathbb{E}_{\tau_w \sim \mathcal{D}}[\pi_\theta(\tau_w|u, \kappa)] \tag{6}$$

Suppose $\mathcal{X} = (x_1, x_2, \ldots, x_{|\mathcal{X}|})$ is the token sequence of the trajectory $\tau_w$, we have:

$$\pi_\theta(\tau_w|u, \kappa) = -\sum_{j=1}^{|\mathcal{X}|} \left( \mathbb{1}(x_j \in \mathcal{A}) \times \log \pi_\theta(x_j|u, \kappa, x_{<j}) \right). \tag{7}$$

Here $\mathbb{1}(x_j \in \mathcal{A})$ is the indicator function to mask tokens unrelated to actions. Please note that $\tau_w$ here **does not include** the state knowledge mentioned in §3.2.

---

[4]For details on how to collect expert trajectories, please refer to Appendix A.

**World Knowledge Model Training.** The main difference in the training data between the agent and knowledge model is **the added state knowledge**. Given the expert trajectories dataset with both task and state knowledge $\mathcal{D}' = \{(u, \kappa, \tau'_w)^{(i)}\}_{i=1}^{|\mathcal{D}'|}$ where $\tau'_w = (a_0, o_0, s_0, \ldots, a_n, o_n, s_n)$, the loss of the knowledge model $\pi_\phi$ can be formulated as:

$$\mathcal{L}_{\text{know}}(\pi_\phi) = -\mathbb{E}_{\kappa, \tau'_w \sim \mathcal{D}'}[\pi_\phi(\kappa|u)\pi_\phi(\tau'_w|u, \kappa)] \tag{8}$$

Suppose $\mathcal{X}' = (x'_1, x'_2, \ldots, x'_{|\mathcal{X}'|})$ is the token sequence of the expert trajectory with state knowledge $\tau'_w$ and $\mathcal{Y} = (y_1, y_2, \ldots, y_{|\mathcal{Y}|})$ represents the token sequence of the task knowledge $\kappa$, we have:

$$\pi_\phi(\kappa|u) = -\sum_{i=1}^{|\mathcal{Y}|} \log \pi_\phi(y_i|u, y_{<i}) \tag{9}$$

$$\pi_\phi(\tau'_w|u, \kappa) = -\sum_{j=1}^{|\mathcal{X}'|} \left(\mathbb{1}(x'_j \in \mathcal{S}) \times \log \pi_\phi(x'_j|u, \kappa, x'_{<j})\right), \tag{10}$$

where $\mathbb{1}(x_j \in \mathcal{S})$ is the indicator function to mask tokens unrelated to state knowledge.

## 3.4 Agent Planning with World Knowledge Model

At inference time, the agent model plans on the evaluation tasks with the aid of the world knowledge model. We redefine the historical trajectory $h_t = (u, \kappa, a_0, o_0, a_1, o_1, \ldots, a_t, o_t)$. Given a specific task instruction $u$, the knowledge model first generates the task knowledge $\kappa \sim \pi_\phi(\cdot|u)$, then the agent model starts planning. Assuming the available action set $\mathcal{A}_u \subseteq \mathcal{A}$ for the task $u$ is $(\alpha_u^{(1)}, \alpha_u^{(2)}, \ldots, \alpha_u^{(|\mathcal{A}_u|)})$, at any time $t \geq 0$, instead of directly generating a next action $a_{t+1} \in \mathcal{A}_u$ based on $h_t$, we first employ the world knowledge model to generate the current state knowledge $s_t \sim \pi_\phi(\cdot|h_t)$ and leverage $s_t$ to query the state knowledge base $\mathcal{B} = \{(s, a_{\text{pre}}, a_{\text{next}})^{(i)}\}_{i=1}^{|\mathcal{B}|}$. With the state knowledge as the key, we retrieve $\mathcal{N}$ nearest triplets **from where** $a_{\text{pre}} = a_t$ based on semantic similarity and collect the corresponding next actions $a_{\text{next}}$. We count the probability of each action $p_{\text{know}}(\alpha_u^{(i)}) = \frac{\mathcal{N}_i}{\mathcal{N}}$, where $\mathcal{N}_i$ is the occurrence number of action $\alpha_u^{(i)}$ in all the collected $a_{\text{next}}$. Therefore, we get the probability acquired from the state knowledge base:

$$P_{\text{know}}(\mathcal{A}_u) = (p_{\text{know}}(\alpha_u^{(1)}), p_{\text{know}}(\alpha_u^{(2)}), \cdots, p_{\text{know}}(\alpha_u^{(|\mathcal{A}_u|)})), \quad \sum_{i=1}^{|\mathcal{A}_u|} p_{\text{know}}(\alpha_u^{(i)}) = 1. \tag{11}$$

Afterward, we sample the probability distribution of the first token for each action $\alpha_u^{(i)}, 1 \leq i \leq |\mathcal{A}_u|$ from the last layer of the agent model and apply a `softmax` function to normalize the probability distribution. We define the probability acquired from the agent model as:

$$P_{\text{agent}}(\mathcal{A}_u) = (p_{\text{agent}}(\alpha_u^{(1)}), p_{\text{agent}}(\alpha_u^{(2)}), \cdots, p_{\text{agent}}(\alpha_u^{(|\mathcal{A}_u|)})), \quad \sum_{i=1}^{|\mathcal{A}_u|} p_{\text{agent}}(\alpha_u^{(i)}) = 1. \tag{12}$$

Finally, we determine the next action by combining the above two probabilities:

$$a_{t+1} = \underset{\alpha_u^{(i)} \in \mathcal{A}_u, 1 \leq i \leq |\mathcal{A}_u|}{\arg\max} (\gamma \cdot p_{\text{agent}}(\alpha_u^{(i)}) + (1 - \gamma) \cdot p_{\text{know}}(\alpha_u^{(i)})), \tag{13}$$

where $\gamma$ is the hyperparameter that controls the proportion of $P_{\text{agent}}(\mathcal{A}_u)$. Based on the above, we enhance the agent planning by global guidance from task knowledge and local constraints from state knowledge generated by our WKM. Due to the WKM and retrieval, the inference stage incurs additional time overhead compared to the pure agent model. The approximate ratio is around 2.5:1.

# 4 Experiments

## 4.1 Experimental Settings

**Datasets and Metrics.** We evaluate our method on three real-world simulated planning datasets: **ALFWorld** [41], **WebShop** [53], and **ScienceWorld** [50]. AlFWorld and ScienceWorld include

Table 1: **Main Results.** The best results are marked in **bold** and the second-best results are marked with underline. All the prompt-based baselines (◔) are evaluated under one-shot prompting and all the fine-tuning-based baselines (◕) are trained through LoRA. Red represents the changes of WKM relative to the optimal results in the baselines. WKM and agent model are different LoRAs sharing the same backbone.

| Backbone | Method | ALFWorld | | WebShop | ScienceWorld | |
|---|---|---|---|---|---|---|
| | | Seen | Unseen | | Seen | Unseen |
| GPT-3.5-Turbo | ◔ REACT | 8.57 | 5.97 | 44.37 | 15.41 | 13.99 |
| GPT-4 | | 44.29 | 38.05 | 62.76 | 67.32 | 65.09 |
| Mistral-7B | ◔ REACT | 7.86 | 5.22 | 14.63 | 20.72 | 17.65 |
| | ◔ Reflexion | 11.56 | 6.00 | 16.64 | 21.07 | 18.11 |
| | ◕ NAT | 64.43 | 68.96 | 61.01 | 57.12 | 50.79 |
| | ◕ ETO | 66.84 | 71.43 | 64.09 | 58.17 | 51.85 |
| | ◕ KNOWAGENT | 70.44 | 70.72 | 61.28 | 59.32 | 47.24 |
| | **WKM** | **73.57** +3.13 | **76.87** +5.44 | **65.48** +1.39 | **62.12** +2.80 | **53.62** +1.77 |
| Gemma-7B | ◔ REACT | 6.43 | 2.24 | 5.93 | 3.58 | 3.51 |
| | ◔ Reflexion | 7.14 | 2.99 | 7.71 | 4.94 | 3.93 |
| | ◕ NAT | 67.86 | 65.88 | 55.82 | 47.63 | 44.98 |
| | ◕ ETO | 66.43 | 68.66 | 62.67 | 50.44 | 47.84 |
| | ◕ KNOWAGENT | 69.29 | 67.60 | 58.80 | 48.55 | 45.28 |
| | **WKM** | **70.71** +1.42 | **70.40** +1.74 | **63.75** +1.08 | **53.68** +3.24 | **49.24** +1.40 |
| Llama-3-8B | ◔ REACT | 2.86 | 3.73 | 19.32 | 24.76 | 22.66 |
| | ◔ Reflexion | 4.29 | 4.48 | 22.73 | 27.23 | 25.41 |
| | ◕ NAT | 60.71 | 59.70 | 61.60 | 55.24 | 48.76 |
| | ◕ ETO | 64.29 | 64.18 | 64.57 | 57.90 | 52.33 |
| | ◕ KNOWAGENT | 66.71 | 62.69 | 64.40 | 58.67 | 49.18 |
| | **WKM** | **68.57** +1.86 | **65.93** +1.75 | **66.64** +2.07 | **60.12** +1.55 | **54.75** +2.42 |

unseen tasks to evaluate the agent's generalization ability. The reward of ALFWorld is binary 0 or 1, indicating whether the agent has completed the task or not. WebShop and ScienceWorld provide dense rewards from 0 to 1 to measure the completion level of the task. For all the datasets, we apply **average reward** as the final metrics. Please refer to Appendix B for detailed dataset information.

**Models and Baselines.** We evaluate on three state-of-the-art open-source models: 1) **Mistral-7B** [16], the Mistral-7B-Instruct-v0.2 version. 2) **Gemma-7B** [24], the Gemma-1.1-7B-it version. 3) **Llama-3-8B** [25], the Meta-Llama-3-8B-Instruct version. We compare our method with two prompt-based baselines: **REACT** [54] and **Reflexion** [40]. Besides, we adopt two strong baselines that introduce rejected trajectories into the training process to learn from experience: **NAT** [49], learn from rejected trajectories through SFT, and **ETO** [44], learn from rejected trajectories through DPO [36]. Moreover, we compare with a knowledge-augmented planning method **KNOWAGENT**. We also include **ChatGPT** (gpt-3.5-turbo-0125) [27] and **GPT-4** (gpt-4-32K-0613) [28] for comparison. All the prompt-based baselines are tested under one-shot and all the fine-tuning-based baselines are trained with LoRA [12]. Please refer to Appendix C for baselines and re-producing details.

**Training and Inference Setups.** We fine-tune the proposed approach with LoRA [12] using the LlamaFactory [62] framework. During training, the model is tuned after finishing the entire trajectory rather than each step of action. The learning rate is 1e-4 and the sequence length is 2048 for all the models. The training epoch is 3 and the batch size is 32. We adopt the AdamW optimizer [22] with a cosine learning scheduler. During inference, we apply the embedding layer of WKM as the encoder and use the **cosine similarity** between sentences for retrieval. The number of retrieved action-state-action triplets $\mathcal{N}$ is set to 3000 and the $P_{\text{agent}}(\mathcal{A}_u)$ weight $\gamma$ is set to {0.4, 0.5, 0.7}. All the training and inference experiments are conducted on 8 NVIDIA V100 32G GPUs within 12 hours. Please refer to Appendix D for detailed hyperparameters used in our paper.

### 4.2 Results

**Main Results.** As shown in Table 1, **for prompt-based baselines** on open-source models, both REACT and Reflexion exhibit poor performance, far behind our method and fine-tuning-based baselines on various datasets. GPT-3.5-Turbo performs ordinarily on two datasets other than WebShop, and it even falls behind Mistral-7B and Llama-3-8B's REACT performance on ScienceWorld. However, GPT-4 exhibits strong performance across various datasets. Nevertheless, our approach, through

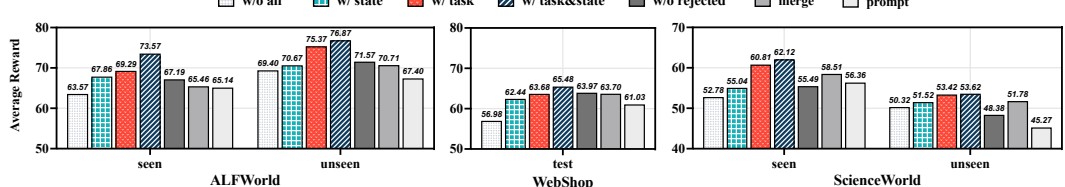

Figure 3: **Ablation Study** on Mistral-7B. **w/o all** means the vanilla experienced agent model training with pure expert trajectories. **w/ state** is testing agent model with only state knowledge base constraints. **w/ task** stands for guiding agent model with only task knowledge. **w/ task&state** is our WKM with both task knowledge guidance and state knowledge constraints. **w/o rejected** means synthesizing task knowledge solely through expert trajectories. **merge** stands for training WKM and the agent model together with one single model. **prompt** means using few-shot prompts to replace the WKM for providing knowledge.

Table 2: **Average Steps.** The maximum number of steps in ALFWorld and WebShop is 40 and 10. In ScienceWorld, the number of steps ranges from 10 to 120 depending on the task type, with an average of around 40.

| Method | ALFWorld | | WebShop | ScienceWorld | |
|---|---|---|---|---|---|
| | Seen | Unseen | | Seen | Unseen |
| NAT | 23.27 | 23.42 | 4.08 | 20.18 | 21.21 |
| ETO | 19.82 | 22.29 | 3.99 | 24.13 | 26.35 |
| KNOWAGENT | 18.51 | 24.56 | 4.01 | 21.06 | 24.74 |
| **WKM** | **17.66** | **17.92** | **3.97** | **18.74** | **19.59** |

Table 3: **Hallucinatory Action Rates** on ALFWorld. We calculate the proportion of trajectories containing invalid actions regardless of their correctness.

| Method | ALFWorld | |
|---|---|---|
| | Seen | Unseen |
| NAT | 45.71% | 50.00% |
| ETO | 34.29% | 36.57% |
| KNOWAGENT | 33.57% | 44.78% |
| **WKM** | **32.86%** | **29.85%** |

LoRA training alone, surpasses GPT-4 on ALFWorld (44.29→73.57 on seen, 38.05→76.87 on unseen) and WebShop (62.76→66.64). **For fine-tuning-based baselines**, both NAT and ETO fall behind our method, implying that just integrating world knowledge for agent models is worth more than further fussy SFT or DPO on negative examples. Our method also performs better than KNOWA-GENT which brings human-designed fixed action knowledge and long action paths into trajectories. This suggests the effectiveness of our WKM which is responsible for generating instance-level task knowledge and maintaining implicit action constraints. Furthermore, KNOWAGENT's performance on unseen tasks is not as impressive as on seen tasks, while WKM can keep its advantage. This phenomenon also demonstrates the generalization ability of WKM.

**Approach Ablations.** As shown in Figure 3, taking Mistral-7B as an example, we decompose the key components of WKM to examine the roles of the task and state knowledge separately. In a macro view, removing each module results in a clear drop in the agent's performance, which validates the power of our world knowledge. Furthermore, the improvement through task knowledge (*w/ task*) is more pronounced than that through state knowledge (*w/ state*), suggesting the necessity of global prior knowledge for agent planning. A more micro observation reveals that the impact of state knowledge is more significant on seen tasks compared to unseen tasks, while the influence of task knowledge is sustainable across seen and unseen tasks. This may be attributed that although our real-time state knowledge is generated by WKM, the state knowledge base is built on the training set, which may weaken generalization to some extent. Additionally, to validate our motivation of allowing the agent to learn task knowledge from both expert and generated trajectories, we exclude the rejected trajectories during the synthesis of task knowledge, instructing the agent model to synthesize knowledge solely based on the chosen trajectories. The results (*w/o rejected*) demonstrate that learning from the contrast between chosen and rejected trajectories is more effective than learning from chosen examples alone. This procedure is a little similar to DPO, but we achieve it through knowledge augmentation rather than directly converting it into a loss calculation between chosen and rejected trajectories. Additional results can further evident that training a WKM separately performs better than training one single model together with the agent model as well as using few-shot prompts to replace WKM for providing knowledge.

### 4.3 Analysis

**World knowledge can mitigate blind trial-and-error and reduce hallucinatory actions.** We compare the number of planning steps for each dataset between three strong baselines and WKM and calculate the average steps of each method. As depicted in Figure 9 (in Appendix F), WKM

demonstrates the ability to complete a significant proportion of tasks using the shortest trajectory, indicating that guidance from world knowledge can effectively reduce the agent's blind trial-and-error in the environment. Taking a further perspective from an average standpoint in Table 2, it can be observed that WKM exhibits lower average planning steps compared to other baselines. As ALFWorld can respond to invalid actions, in Table 3, we count the percentage of hallucinatory actions that occurred in trajectories from ALFWorld for each method. The results confirm the effectiveness of our world knowledge model to decrease hallucinatory actions. Furthermore, it is worth noting that most baselines show a prominent increase in the average number of steps and percentage of invalid actions when transitioning from seen tasks to unseen tasks, but WKM can still maintain a relatively low level. This reflects laterally that our world knowledge can still effectively guide the agent model on unseen tasks, highlighting the knowledge generalization brought by the world knowledge model. To see how our world knowledge works, please refer to our case study in Appendix H.

**Our instance-level knowledge can generalize better to unseen tasks.** To further explore the benefit of using a knowledge model to generate instance-level task knowledge, we carefully survey the task knowledge generated by our WKM and abstract it into dataset-level knowledge for each dataset. Then we retrain the agent model to adapt to new dataset-level knowledge[5]. As illustrated in Figure 4, we compare the performance of dataset-level knowledge with our instance-level task knowledge (WKM *w/o state*) on ALFWorld and ScienceWorld. It can be observed that our model-generated instance-level knowledge not only surpasses human-designed knowledge on seen tasks but also exhibits even more remarkable performance on unseen tasks, with the improvement in performance on unseen tasks significantly greater than that on seen tasks. This phenomenon straightly reflects the strong generalization ability of our knowledge model compared to rigidly designed knowledge by humans.

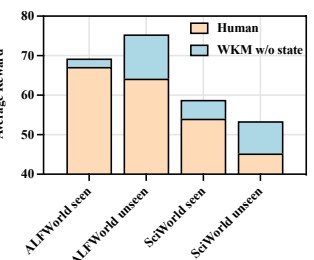

Figure 4: Performance of human-designed dataset-level knowledge compared to WKM generated instance-level knowledge.

Table 4: **Weak-guide-strong**. The knowledge model here is based on Mistral-7B.

| Backbone | Method | ALFWorld | |
|---|---|---|---|
| | | Seen | Unseen |
| GPT-3.5-Turbo | REACT | 8.57 | 5.97 |
| | **WKM** w/o state | **12.86** | **8.96** |
| GPT-4 | REACT | 44.29 | 38.05 |
| | **WKM** w/o state | **50.71** | **47.01** |

**Weak knowledge model guides strong agent model planning.** In our main experiments, the knowledge model and agent model are based on the same backbone. Here, we explore on ALFWorld what will happen if we use a weak knowledge model to guide a strong agent model. We choose Mistral-7B as the backbone of the knowledge model and ChatGPT and GPT-4 as the agent model. Since we cannot get the token distribution from OpenAI API, we only apply task knowledge to the agent model. As exhibited in Table 4, the results of both ChatGPT and GPT-4 show distinct advances after being guided by the Mistral-7B world knowledge model, indicating the weak world knowledge model also contains knowledge that the strong model may lack. In the era of LLMs, this inspires us with a new agent learning paradigm: **weak-guide-strong**. Due to its lightweight nature, the weak knowledge model can flexibly adjust its parameters based on the needs of the agent model, which can address the difficulty of large agent models in adapting to new environments through fine-tuning.

**Unified World Knowledge Model Training.** We mix the world knowledge collected from all three datasets and jointly train one single world knowledge model to investigate the effect of multi-task world knowledge learning. Figure 5 illustrates the relative performance comparison between multi-task WKM and various baselines, from which we can observe that multi-task WKM not only does not lead to performance degradation but also exhibits visible improvements compared to single-task WKM, especially on WebShop and ScienceWorld. Similar to [57, 58, 3] which endeavor to train a unified agent model and achieve strong generalization ability to held-out tasks, this observation inspires us with the potential of training a unified world knowledge model that can be applied to help

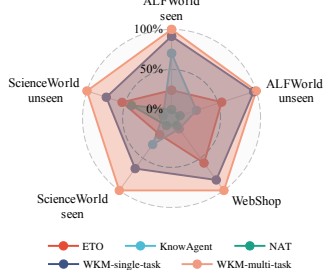

Figure 5: Relative performance of multi-task WKM compared to various baselines.

---

[5]Detailed manually designed dataset-level knowledge prompt can be found in Appendix I.3

various held-in agent models and also generalize to guide held-out agent models. A more daring idea is whether a unified agent model combined with a unified world knowledge model is the key to Artificial General Intelligence (AGI).

**Explicit state knowledge will hurt the planning performance.** To demonstrate the rationality of our choice to construct a state knowledge base, we explore the effect of directly incorporating state knowledge into the context of the agent model (we retrain the agent model to follow both the task and state knowledge), as shown in Figure 6. The performance of explicit state knowledge is far inferior to our approach of retrieving from a state knowledge base and utilizing probabilistic constraints. It even performs worse than

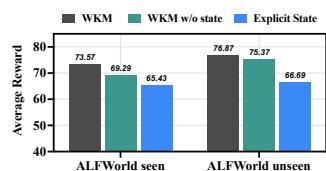

Figure 6: Performance of explicit state knowledge.

when we remove state knowledge and only include task knowledge. This clearly indicates that blindly extending prompts with a large amount of explicit natural language feedback is lose-more-than-gain for agent planning, and implicit knowledge constraints may be sometimes more prudent.

**Case Study.** In Figure 10 (Appendix H), we list the trajectories of ETO and our WKM within the same task in ALFWorld to illustrate how world knowledge functions. **The rationales before each action have been omitted to guarantee a clear illustration.** The task is to `clean some soapbar and put it in cabinet`. Initially, ETO blindly searches for the `soapbar` in the `countertop` and `cabinet`, introducing a lot of irrelevant information and unnecessary context. In the later stages of planning, ETO experiences the hallucination and executes the `put` action after `close` the cabinet, causing the environment to become unrecognizable and resulting in a collapse. On the contrary, guided by task knowledge, WKM directly identified the possible locations of the `soapbar` and successfully found it in the first attempt. Subsequently, WKM efficiently completed the task with precision, adhering to the constraints of state knowledge.

# 5   Related Work

**LLM Agents.** LLMs have emerged as a promising avenue towards unlocking the potential of Artificial General Intelligence, offering robust support for the development of agent systems [48, 51, 8, 63]. Existing works in this field mainly focuses on agent planning [14, 21, 54, 42], external tools harnessing [39, 23, 43, 29, 32, 35, 46], code generation [45, 21, 31, 11], etc. Recently, there has been an increasing focus on endowing open-source LLMs with agent functionalities through fine-tuning [2, 57, 56, 38, 44, 49]. However, these approaches rely on blindly fitting the probabilities of tokens to learn planning, without having an intimate cognition of the environment. The lack of knowledge can lead to the agent blindly attempting trial-and-error and generating hallucinatory actions.

**Knowledge Augmented Agent Planning.** Planning [15] is a crucial capability for intelligent agents to accomplish real-world tasks, often requiring agents to possess rich knowledge and environmental commonsense. Few works have explored the field of knowledge-augmented agent planning. [14, 61, 5] utilize the rich parametric knowledge stored in pre-trained language models to assist agent planners. [7, 20, 59, 64] design structured or natural language knowledge to regulate the actions. However, the above studies require the manual design of fixed prompt templates or task procedures, making it challenging to transfer across different task environments. [63, 55, 6] propose the automation of knowledge generation using language models. However, their knowledge either consists of only global workflow or only local action principles. In contrast, we train our world knowledge model both on global task knowledge and local state knowledge to assist agent planning, and these knowledge sources are derived from the model's self-summary rather than hand-curated.

**LLM-based World Model.** World model and agent model often co-occur in the domain of reinforcement learning and robotics [13, 9, 19, 37, 26, 4]. With LLMs commonly deemed as the most powerful intelligent machines constructed by humans thus far, the LLM-backed world models have been proposed [61, 10, 13]. In our paper, we attempt to self-synthesize world knowledge and train to obtain a world knowledge model. However, we consider our model to be a world **knowledge model** rather than a **world model** based on the reason that our model is temporarily unable to utilize search algorithms (e.g. MCTS) in conjunction with the agent model to make predictions about the world and we leave this for our future work.

# 6 Conclusion and Future Work

In this paper, we strive to develop a parametric world knowledge model (WKM) to augment language agent model planning. Our WKM can generate prior task knowledge to guide global planning as well as dynamic state knowledge to regulate local planning. Our extensive results show that our world knowledge can work on both GPT-4 and state-of-the-art open-source models and achieve superior performance compared to various strong baselines. Analytical experiments validate that our WKM can 1) reduce brainless trial-and-error and invalid actions, 2) generalize better to unseen tasks, 3) achieve weak-guide-strong, and 4) be effectively extended to unified world knowledge training. Potential future directions include: 1) building a unified world knowledge model, 2) learning to predict the world like a world model, 3) applying to multi-modal agent planning, etc.

## Limitations

Despite our best efforts, this paper still has some limitations: 1) Our primary intention behind designing the WKM is to compensate for the lack of world knowledge in the agent model. However, determining what a language model knows and doesn't know has been an ongoing challenge that remains unresolved. 2) It is widely acknowledged that world knowledge extends beyond textual representations. While our world knowledge is currently limited to textual information, exploring multi-modal world knowledge models is indeed one of our important future tasks. 3) Our world knowledge model cannot dynamically update with the changes of the world and feedback from the agent. 4) Generating world knowledge can introduce additional inference overhead.

## Acknowledgments and Disclosure of Funding

We would like to express our great gratitude to the anonymous reviewers for their kind comments. This work was supported by the National Natural Science Foundation of China (No. 62206246, No. NSFCU23B2055, No. NSFCU19B2027), the Fundamental Research Funds for the Central Universities (226-2023-00138), Zhejiang Provincial Natural Science Foundation of China (No. LGG22F030011), Yongjiang Talent Introduction Programme (2021A-156-G), CIPSC-SMP-Zhipu Large Model Cross-Disciplinary Fund, Ningbo Science and Technology Special Projects under Grant No. 2023Z212, Information Technology Center and State Key Lab of CAD&CG, Zhejiang University, NUS-NCS Joint Laboratory (A-0008542-00-00), and the Ministry of Education, Singapore, under the Academic Research Fund Tier 1 (FY2023) (Grant A-8001996-00-00). We gratefully acknowledge the support of Zhejiang University Education Foundation Qizhen Scholar Foundation.

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

## A  Expert Trajectories Collection

We mainly use the expert trajectories with a REACT-style [54] collected from [44]:

1. **ALFWorld** [41]. The dataset provides human-annotated trajectories.

2. **WebShop** [53]. Except for human-annotated trajectories, GPT-4 is also applied to generate trajectories with a reward larger than 0.7 being reserved.

3. **ScienceWorld** [50]. The dataset offers heuristic algorithms to search golden trajectories for each sub-task.

Since the original golden trajectories do not contain rationales, GPT-4 is further leveraged to generate the corresponding information.

## B  Dataset Information

We evaluate our method on three real-world simulated agent planning datasets: ALFWorld [41], WebShop [53], and ScienceWorld [50].

1. **ALFWorld** is a household dataset requiring the agent to navigate through the room and manipulate objects. Except for seen tasks, AlFWorld also includes unseen tasks to evaluate the agent's generalization ability. The reward of ALFWorld is binary 0 or 1, indicating whether the agent has completed the task or not.

2. **WebShop** is an online shopping dataset in a website environment. It provides dense final rewards from 0 to 1 to measure the completion level of the task.

3. **ScienceWorld** is a scientific reasoning dataset which is at the level of a standard elementary school science curriculum. It also possesses both seen and unseen parts and a dense reward function from 0 to 1.

For all the datasets, we apply **average reward** as the final metrics. Table 5 illustrates the statistics of each dataset.

Table 5: Dataset statistics.

| Dataset | Train | Text-Seen | Text-Unseen |
|---|---|---|---|
| ALFWorld | 3,119 | 140 | 134 |
| WebShop | 1,824 | 200 | - |
| ScienceWorld | 1,483 | 194 | 211 |

## C  Compared Baselines

Here we detailedly introduce the baselines we compare with and our re-produce details.

1. **REACT** [54]. The first approach incorporates Chain-of-Thought (COT) prompting in agent planning tasks with a format of Thought-Action-Observation loop. In our paper, we apply one-shot prompting for REACT[6].

2. **Reflexion** [40]. A strong prompt-based baseline reinforces agent planning with verbal feedback. Manually designed prompts are used to enable the agent to reflect on the historical trajectory and re-plan based on the feedback. In our paper, we utilize one-shot prompting for reflection and select the first reflect iteration as our result due to limited context[7].

3. **NAT** [49]. NAT includes negative trajectories by employing different prompts during agent fine-tuning. When evaluating, only positive prompts are used to encourage the language

---

[6] https://github.com/ysymyth/ReAct
[7] https://github.com/noahshinn/reflexion

agent to generate correct trajectories. As it also follows the REACT-style format, we directly use the default positive and negative prompts and train with LoRA in our paper[8].

4. **ETO** [44]. Another baseline includes negative trajectories during agent training. The method contains two training phases, of which the first phase is behavior cloning which fine-tunes the agent on expert trajectories, and the second phase is learning from failures which further fine-tunes the agent through Direct Preference Optimization (DPO) [36]. In our paper, we remove the one-shot prompt for fairness and retain all the default hyperparameters proposed in ETO except for LoRA training[9].

5. **KNOWAGENT** [64]. KNOWAGENT is a knowledge-augmented agent planning baseline that applies action knowledge in the prompt and maintains an action path in the context during planning to constrain the agent's action. We directly use the default prompt mentioned in KNOWAGENT for ALFWorld and carefully extend it to WebShop and ScienceWorld by following a similar format[10].

All the prompt-based baselines are tested under one-shot and all the fine-tuning-based baselines are trained with LoRA [12].

# D  Hyperparameters

The detailed hyperparameters we use during training and inference are shown in Table 6. We employ identical hyperparameters for different models. The temperature of the agent model is set to 0.0 when conducting exploration and 0.5 when introduced into WKM. The temperature of WKM is set to 0.0 for all the time. The $P_{\text{agent}}(\mathcal{A}_u)$ weight $\gamma$ is set to 0.4 for ALFWorld, 0.5 for WebShop, and 0.7 for SienceWorld.

Table 6: Detailed hyperparameters used in our paper.

| Name | Value |
|---|---|
| lora r | 8 |
| lora alpha | 16 |
| lora dropout | 0.05 |
| lora target modules | q_proj, v_proj |
| cutoff len | 2048 |
| epochs | 3 |
| batch size | 32 |
| batch size per device | 4 |
| gradient accumulation steps | 2 |
| learning rate | 1e-4 |
| warmup ratio | 0.03 |
| temperature | 0.0, 0.5 |
| retrieved state knowledge $\mathcal{N}$ | 3000 |
| $P_{\text{agent}}(\mathcal{A}_u)$ weight $\gamma$ | 0.4, 0.5, 0.7 |

# E  Training Data Example

We show the training data example for agent model and world knowledge model in Figure 7 and Figure 8. During training, transparent tokens do not contribute to the calculation of the loss. Only the parts of the model responsible for output (agent model: purple, knowledge world model: green), participate in the loss calculation.

---

[8]https://github.com/Reason-Wang/NAT
[9]https://github.com/Yifan-Song793/ETO
[10]https://github.com/zjunlp/KnowAgent

## Training Data Example for Agent Model

**Task Instruction:** You are in the middle of a room. Looking quickly around you, you see a bed 1, a desk 1, a drawer 5, a drawer 4, a drawer 3, a drawer 2, a drawer 1, a dresser 1, a garbagecan 1, a laundryhamper 1, and a shelf 1. Your task is to: examine the book with the desklamp.

**Task Knowledge:** When you cannot find the object required for the task under the desklamp, first check all the drawers and shelves in the room, then revisit the previous locations and double-check before giving up and looking in the garbage can or laundry hamper. The action workflows are as follows: 1. For each drawer and shelf, go to the location and open it to check if the object is inside. 2. If the object is not found, close the drawer or shelf and move on to the next one. 3. After checking all the drawers and shelves, revisit the previous locations and double-check if the object might have been overlooked. 4. If the object is still not found, look in the garbage can or laundry hamper as a last resort.

**Thought:** The task requires me to examine a book with the desklamp. First, I need to locate the book.

**Action:** go to bed 1

**Observation:** On the bed 1, you see a book 1, a laptop 1, and a pillow 1.

**Thought:** I see a book on the bed. I should pick it up to examine it with the desklamp.

**Action:** take book 1 from bed 1

**Observation:** You pick up the book 1 from the bed 1.

**Thought:** Now that I have the book, I need to find the desklamp to examine the book under its light.

**Action:** go to desk 1

**Observation:** On the desk 1, you see an alarm clock 1, a bowl 1, a CD 2, a mug 3, a mug 2, a mug 1, a pen 1, and a pencil 2.

**Thought:** The desklamp isn't on the desk. I need to look around the room to find where the desklamp is located.

**Action:** go to dresser 1

**Observation:** On the dresser 1, you see a cellphone 1, a desklamp 1, and a keychain 2.

**Thought:** I've found the desklamp on the dresser. I should use it to examine the book.

**Action:** use desklamp 1

Figure 7: Training Data Example for Agent Model.

## Training Data Example for World Knowledge Model

**Task Instruction:** You are in the middle of a room. Looking quickly around you, you see a bed 1, a desk 1, a drawer 5, a drawer 4, a drawer 3, a drawer 2, a drawer 1, a dresser 1, a garbagecan 1, a laundryhamper 1, and a shelf 1. Your task is to: examine the book with the desklamp.

**Task Knowledge:** When you cannot find the object required for the task under the desklamp, first check all the drawers and shelves in the room, then revisit the previous locations and double-check before giving up and looking in the garbage can or laundry hamper. The action workflows are as follows: 1. For each drawer and shelf, go to the location and open it to check if the object is inside. 2. If the object is not found, close the drawer or shelf and move on to the next one. 3. After checking all the drawers and shelves, revisit the previous locations and double-check if the object might have been overlooked. 4. If the object is still not found, look in the garbage can or laundry hamper as a last resort.

**Thought:** The task requires me to examine a book with the desklamp. First, I need to locate the book.

**Action:** go to bed 1

**Observation:** On the bed 1, you see a book 1, a laptop 1, and a pillow 1.

**State Knowledge:** Your task is to examine a book with the desklamp. You have not found the desklamp yet, and have only checked the bed 1.

**Thought:** I see a book on the bed. I should pick it up to examine it with the desklamp.

**Action:** take book 1 from bed 1

> **Observation:** You pick up the book 1 from the bed 1.
> **State Knowledge:** Your task is to examine a book using the desklamp. Your have located the book on bed 1 and picked it up.
> **Thought:** Now that I have the book, I need to find the desklamp to examine the book under its light.
> **Action:** go to desk 1
> **Observation:** On the desk 1, you see an alarm clock 1, a bowl 1, a CD 2, a mug 3, a mug 2, a mug 1, a pen 1, and a pencil 2.
> **State Knowledge:** Your task is to examine a book with the desklamp. You have located the book on bed 1 and picked it up. Next, You need to find the desklamp to examine the book under its light.
> **Thought:** The desklamp isn't on the desk. I need to look around the room to find where the desklamp is located.
> **Action:** go to dresser 1
> **Observation:** On the dresser 1, you see a cellphone 1, a desklamp 1, and a keychain 2.
> **State Knowledge:** Your task is to examine a book with the desklamp. You have located the book on the bed and picked it up, now you find a desklamp on a dresser.

Figure 8: Training Data Example for World Knowledge Model.

## F  Win Rate of Planning Steps

See Figure 9.

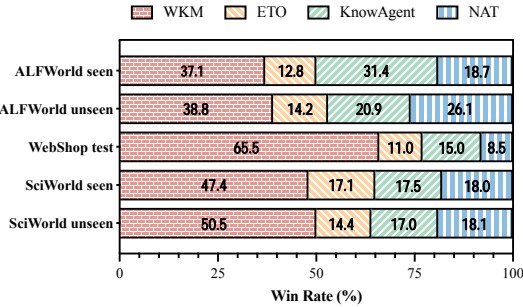

Figure 9: **Win Rate of Planning Steps.** We choose the method with the shortest steps for each task and calculate the proportion.

## G  Impact of $\gamma$

In fact, the ratio $\gamma$ can be viewed as a signal to reflect whether knowledge or planning is more important for a task. To understand which part of the output action has the most significant impact, we further analyze $\gamma = 0$ (fully trust state knowledge base) and $\gamma = 1$ (fully trust agent model, equivalent to remove state knowledge in Figure 3). The empirical results can be seen in Table 7. It can be observed that state knowledge primarily serves as a constraint to alleviate hallucinated actions for the agent model. However, when we fully trust it ($\gamma = 0$), its lack of generalization significantly harms the performance of the agent model.

## H  Case Study

In Figure 10, we list the trajectories of ETO and our WKM within the same task in ALFWorld to illustrate how world knowledge functions. **The rationales before each action have been omitted to guarantee a clear illustration.** The task is to clean some soapbar and put it in cabinet. Initially, ETO blindly searches for the soapbar in the countertop and cabinet, introducing a

Table 7: **Impact of** $\gamma$. In fact, the ratio $\gamma$ can be viewed as a signal to reflect whether knowledge or planning is more important for a task. To understand which part of the output action has the most significant impact, we further analyze $\gamma = 0$ (fully trust state knowledge base) and $\gamma = 1$ (fully trust agent model, equivalent to remove state knowledge in Figure 3). It can be observed that state knowledge primarily serves as a constraint to alleviate hallucinated actions for the agent model. However, when we fully trust it ($\gamma = 0$), its lack of generalization significantly harms the performance of the agent model.

| Method | ALFWorld | | WebShop | ScienceWorld | |
|---|---|---|---|---|---|
| | Seen | Unseen | | Seen | Unseen |
| $\gamma = 0$ | 1.58 | 0.00 | 25.83 | 18.69 | 15.37 |
| $\gamma = 1$ | 69.29 | 75.37 | 63.68 | 60.81 | 53.42 |
| **WKM** | **73.57** | **76.87** | **65.48** | **62.12** | **53.62** |

lot of irrelevant information and unnecessary context. In the later stages of planning, ETO experiences the hallucination and executes the `put` action after `close` the cabinet, causing the environment to become unrecognizable and resulting in a collapse. On the contrary, guided by task knowledge, WKM directly identified the possible locations of the `soapbar` and successfully found it in the first attempt. Subsequently, WKM efficiently completed the task with precision, adhering to the constraints of state knowledge.

# I Prompts

In this section, we illustrate all the prompts used in our paper. Part of our prompts are taken from AutoGuide [6].

## I.1 Task Knowledge Synthesis Prompt

See Figure 11.

## I.2 State Knowledge Summarization Prompt

See Figure 12.

## I.3 Dataset-Level Knowledge Prompt

See Figure 13.

# J Ethics Statement

This research was conducted following the ethical standards and best practices. All our experiments use publicly available datasets (as detailed in Appendix B), avoiding ethical concerns related to privacy, confidentiality, or misuse of personal biological information. However, despite our best efforts, it is not avoidable if someone maliciously modifies the world knowledge model to contradict the world's knowledge and leads the agent to engage in unethical behavior.

## Case Study of WKM

### ETO

**Task Instruction:** You are in the middle of a room. Looking quickly around you, you see a cabinet 4, a cabinet 3, a cabinet 2, a cabinet 1, a countertop 1, a garbagecan 1, a handtowelholder 2, a handtowelholder 1, a sinkbasin 2, a sinkbasin 1, a toilet 1, a toiletpaperhanger 1, and a towelholder 1. Your task is to: clean some soapbar and put it in cabinet.
*# Blind trial-and-error begins.*
**Agent:** go to countertop 1
**Observation:** On the countertop 1, you see a candle 3, a candle 1, a soapbottle 1, a spraybottle 2, and a toiletpaper 2.
**Agent:** go to cabinet 1
**Observation:** The cabinet 1 is closed.
**Agent:** open cabinet 1
**Observation:** You open the cabinet 1. The cabinet 1 is open. In it, you see a soapbottle 2.
**Agent:** close cabinet 1
**Observation:** You close the cabinet 1.
*# Blind trial-and-error ends.*
**Agent:** go to toilet 1
**Observation:** On the toilet 1, you see a soapbar 1, and a spraybottle 3.
**Agent:** take soapbar 1 from toilet 1
**Observation:** You pick up the soapbar 1 from the toilet 1.
**Agent:** go to sinkbasin 1
**Observation:** On the sinkbasin 1, you see nothing.
**Agent:** clean soapbar 1 with sinkbasin 1
**Observation:** You clean the soapbar 1 using the sinkbasin 1.
**Agent:** go to cabinet 1
**Observation:** The cabinet 1 is closed.
**Agent:** open cabinet 1
**Observation:** You open the cabinet 1. The cabinet 1 is open. In it, you see a soapbottle 2.
**Agent:** close cabinet 1
**Observation:** You close the cabinet 1.
*# Hallucinatory action.*
**Agent:** put soapbar 1 in/on cabinet 1
**Observation:** Nothing happens.
**Agent:** check cabinet 1
**Observation:** Nothing happens.
**Agent:** go to exit
**Observation:** Nothing happens.
**Agent:** exit
**Observation:** Nothing happens.
......
**Reward:** 0.0

### WKM

**Task Instruction:** ...
**Task Knowledge:** When the task is to clean and place an object, you should first locate the object. The possible locations of the soapbar are: toilet, sinbasin. Once the object is found, you should clean it using the sinkbasin, then put it in the cabinet. The action workflows are as follows: 1. Go to the place where the object is located. 2. Take the object from the place. 3. Clean the object using the sinkbasin. 4. Go to the cabinet. 5. Put the object in/on the cabinet.
**Agent:** go to toilet 1
**Observation:** On the toilet 1, you see a soapbar 1, and a spraybottle 3.
**State Knowledge:** Your task is to clean some soapbar and put it in cabinet. Now you are checking toilet 1 and there is a soapbar 1.
**Agent:** take soapbar 1 from toilet 1
**Observation:** You pick up the soapbar 1 from the toilet 1.
**State Knowledge:** Your task is to clean some soapbar and put it in cabinet. You have checked the toilet 1 and found the soapbar 1.
**Agent:** go to sinkbasin 1
**Observation:** On the sinkbasin 1, you see nothing.
**State Knowledge:** Your task is to clean some soapbar and put it in cabinet. You have checked the toilet 1 and found the soapbar 1. Now you are at the sinkbasin.
**Agent:** clean soapbar 1 with sinkbasin 1
**Observation:** You clean the soapbar 1 using the sinkbasin 1.
**State Knowledge:** Your task is to clean some soapbar and put it in cabinet. You have cleaned the soapbar 1 using the sinkbasin 1.
**Agent:** go to cabinet 1
**Observation:** The cabinet 1 is closed.
**State Knowledge:** Your task is to clean some soapbar and put it in cabinet. You have cleaned the soapbar 1 using the sinkbasin 1. Now you are checking cabinet 1.
**Agent:** open cabinet 1
**Observation:** You open the cabinet 1. In it, you see a soapbottle 2.
**State Knowledge:** Your task is to clean some soapbar and put it in cabinet. You have cleaned the soapbar 1 and opened the cabinet 1.
**Agent:** put soapbar 1 in/on cabinet 1
**Reward:** 1.0

Figure 10: Case study for our WKM.

---

**Prompt for Task Knowledge Synthesis**

**Task Knowledge**

**Prompt for Synthesis:** I will provide you with an analysis of both a successful trajectory and an explored trajectory for the same task. By comparing the two, we can identify the key factors that contribute to success. Based on this analysis, you need to generate task-related task knowledge to help increase the success rate of future endeavors.
Success Trajectory: **Success_T**
Explored Trajectory: **Explored_T**
The task knowledge should specify what to do in what task. Here is a task knowledge example:
**Task Knowledge Example**
You should make your answer concise. Put your answer in this format: Task Knowledge: When ... you should (or should not) ... The action workflows are: ...

---

Figure 11: Prompt for Task Knowledge Synthesis.

---

**Prompt for State Knowledge Synthesis**

**State Knowledge**

**Prompt for Synthesis:** You'll get a segment of a trajectory of a text-based task. Your task is to generate a brief and general state knowledge of the task state now, following "State Knowledge: ". Keep it wise and general for the same task. Here is an example:
**State Knowledge Example**
Now it's your turn. Here is the trajectory :
**Trajectory**
Make sure your output is within 128 tokens.
Put your answer in this format: State Knowledge: . . .

---

Figure 12: Prompt for State Knowledge Summarization.

> **Task Knowledge example**
>
> ### Alfworld Task Knowledge example
>
> When picking an object, heat it, and place it, you should first go to the possible locations of the object, then take the object, heat it with microwave, and put it in place.
> The action workflows are as follows:
> 1) go to receptacle
> 2) take object from receptacle
> 3) heat object with receptacle
> 4) go to the place to put the object
> 5) put object in/on receptacle
>
> ### Webshop Task Knowledge example
>
> When looking for an object you want to buy, you should first search with relevant keywords tailored to the product you are looking for, and then click the relevant tag to view the product details, if the description matches the characteristics of the target item, click[buy now].
> The action workflows are as follows:
> 1) search with keywords or examples, if you are searching for a laptop, you might search[laptop, 14-inch, Intel Core i7]
> 2) click the most relevant tag to view the detailed product page.
> 3) check the product details one by one, like color, size, type, and price, and make sure the price is within budget.
> 4) if find the right items, click[buy now] to buy it.
>
> ### Sciworld Task Knowledge example
>
> When tasked with boiling apple juice, focus on locating the kitchen first. Then, locate the apple juice in the fridge. Activate the stove, pour the apple juice into a metal pot, and move the metal pot to the stove. Monitor the stove until the apple juice reaches a boiling point. Once boiled, remove the pot from the stove.
> The action workflows are:
> 1) teleport to the kitchen.
> 2) look around to find the apple juice in the fridge.
> 3) activate the stove.
> 4) pour apple juice into a metal pot.
> 5) move the metal pot to the stove.
> 6) look at stove.
> 7) examine apple juice to confirm boiling.
> 8) repeat step 6,7 until apple juice is boiled.

Figure 13: Dataset-Level Task Knowledge Examples.

