# OpenReview forum: "Agent Planning with World Knowledge Model"
_NeurIPS.cc/2024/Conference — NeurIPS 2024 poster_

### Official Review · Reviewer_8XFL · 2024-06-13

**Soundness:** 3
**Presentation:** 4
**Contribution:** 3
**Rating:** 6
**Confidence:** 4

**Summary:**

This paper presents a parametric world knowledge model designed to enhance agent planning. The model synthesizes knowledge from expert and sampled trajectories for training purposes. It incorporates prior task knowledge for global planning and dynamic state knowledge for local planning. The implementation shows improved performance over several robust baselines using open-source LLMs such as Mistral-7B and Gemma-7B. Incorporating a world knowledge model into LLM-based agents for planning purposes is a novel approach that also helps mitigate common issues like hallucinations and invalid actions in language agents.

**Strengths:**

The paper is overall well-written and the method is novel. It also show promising results using only 7B models that could barely perform planning with basic methods. It's also quite reasonable to introduce a world knowledge model to enhance task specific knowledge during planning.

**Weaknesses:**

Here are some additional points that I believe could be improved:

1. Clarity on hyperparameters and settings: Is the WKM tuned after each step of action or finishing an entire trial? Also what is the split for seen/unseen task? Some tasks are very similar in Alfworld and Scienceworld. What is the structure of the retriever? What is the number of generations for each action?

2. Is WKM training more important for commmon-sense intensive tasks like Scienceworld? Have you previously tested on pure planning tasks? e.g. Blocksworlds, Game 24.  Also, does the ratio $\gamma$ reflects whether knowledge or planning is more important for a task?

3. What is the additional computation overhead for inferencing using a WKM? Also beam search is necessary to obtain more than one action, what is the computation time compared to ReAct?

4. What is the difference between state knowledge and thoughts/ reflection? It seems to be also interleaved between actions. Does the content of state knowledge greatly affect the performance? e.g. using the WKM to enhance ReAct thoughts. The state-knowledge examples in Figure 8 seems more similar to thoughts rather than world knowledge, i.e. give a general commonsense about the environment.

5. Can WKM and agent training be merged together? i.e. using a single model to learn both knowledge and trajectory and update the loss together.

6. Can this method be applied to an online setting? Where knowledge base is directly updated after each step of action so that later actions could use results from exploration.

**Questions:**

Please refer to weaknes.

**Limitations:**

Yes.

---

> ### Author Rebuttal · Authors · 2024-08-04
>
> We are deeply grateful for your valuable time and insightful feedback. Below are our detailed responses to your concerns.
>
> **Q1: Clarity on hyperparameters and settings**
>
> We sincerely apologize for any confusion caused by the details not clarified in the paper.
>
> (1) **As shown in Eqn 8-10**, our WKM is fine-tuned over the entire trajectory, where the loss calculation only involves knowledge, and other irrelevant tokens will be masked.
>
> (2) The seen and unseen tasks are predefined within the dataset. For the unseen tasks, the model has not previously encountered the task types in the training set, requiring the application of more extensive generalization capabilities to address them.
>
> (3) Our retrieval is primarily based on cosine similarity. The encoder is WKM's own embedding layer (for example, the embedding layer of Mistral-7B), and similarity is calculated based on the cosine similarity between sentences.
>
> (4) At each step, we only generate an action once. Since we use open-source models, we can directly obtain the probability distribution of each action's first token from the last layer of the agent model at each step. **As we mentioned in lines 162-163**, we normalize the probability distribution of the action tokens with a softmax function to get the final action distribution $P_{\rm agent}(\mathcal{A}_u)$ from the agent model.
>
> **Q2: Is WKM training more important for commonsense intensive tasks like Sciworld?**
>
> In this paper, we focus more on embodied planning tasks that interact with the environment, which require world knowledge to be resolved. **Tasks such as Blocksworld and Game24 are, in our understanding, more like reasoning tasks**, where the model only needs to reason based on the problem without the need for the environment.
>
> Regarding the ratio $\gamma$ reflecting whether knowledge or planning is more important for a task, your understanding is very insightful. We have also found that for tasks in SciWorld that require a large amount of domain knowledge, the value of $\gamma$ is smaller (knowledge occupies a larger proportion).
>
> **Q3: The additional computation overhead for inferencing using a WKM**
>
> **In fact in lines 167-168, we have discussed the issue of computation overhead.** When accounting for the inference time of the knowledge model and the retrieval time, our total inference time is 2.5 times that of a pure agent model fine-tuned with expert trajectories. In Q1, we explained that we do not employ beam search, and because ReAct is a prompt-based method, which tends to have poorer performance and thus results in significantly longer trajectory lengths, the ratio of our method's inference time compared to ReAct will be even smaller.
>
> **Q4: The difference between state knowledge and thoughts/reflection**
>
> The original intention behind the design of state knowledge is due to our observation that a significant issue in agent planning arises when the trajectory is lengthy, causing the model to experience hallucinations due to difficulties in processing long contexts. Thus, state knowledge acts more like an "outside" prompter, providing the agent with dynamic knowledge of the current environment rather than prior commensense knowledge, to make it focus more on the current action. In contrast, the agent's thoughts resemble an "inside" judger of the current state, which can easily be influenced by long contexts and deviate from the correct path.
>
> **Q5: Can WKM and agent training be merged together?**
>
> We train WKM and the agent model separately for two reasons: 1) Numerous studies have indicated that the division of labor and collaboration among models can lead to the emergence of group intelligence, which achieves better results than individual intelligence; 2) Decoupling the WKM from the agent model allows for more flexible expansion, such as guiding a stronger agent model with a smaller WKM (Table 4), and realizing a unified WKM (Figure 5). To fully show the advantages of separate training, we further conduct experiments where a single model learns planning and knowledge at the same time:
>
> | Mistral-7B   | ALFWorld seen | ALFWorld unseen | WebShop | SciWorld seen | SciWorld unseen |
> | ------------ | ------------- | --------------- | ------- | ------------- | --------------- |
> | single-model | 65.46         | 70.71           | 63.70   | 58.51         | 51.78           |
> | WKM          | 73.57         | 76.87           | 65.48   | 62.12         | 53.62           |
>
> The experiment has shown that the merging of WKM and agent model results in a worse effect, which also confirms our viewpoint.
>
> **Q6: Can this method be applied to an online setting?**
>
> In fact, our initial plan was to create an online version of WKM, but we encountered the following challenges:
>
> 1) **Dynamic adjustment of $\gamma$**. In the early stages, when the state knowledge base contains too few samples to cover common scenarios, we obviously cannot trust its probabilities, so we need to set $\gamma$ to a high value. Conversely, when the state knowledge base is sufficiently populated, $\gamma$ should be set to a relatively lower value. Thus, $\gamma$ should gradually decrease and eventually stabilize over the process, but controlling the speed of decrease and the time to reach stability is challenging.
>
> 2) **Parameter updating of the WKM**. How to adjust the parameters of WKM when new knowledge is acquired involves the frequency of parameter updates. An excessively high frequency could lead to extremely low efficiency, while an insufficient frequency may not meet the needs of the agent model.
>
> We are also conducting further research on the online version of WKM and hope to make significant progress in the future.
>
> Thank you again for your constructive suggestions!
>
> **Please let us know if you have any further questions. If you find that our response addresses some of your concerns, would you kindly consider raising your rating score for our paper? We greatly appreciate your consideration.**

---

> > ### Comment · Reviewer_8XFL · 2024-08-14
> > **Thanks for the rebuttal**
> >
> > Your rebuttal has clearly addressed my concerns and helped me gain a better understanding of world knowledge training. I have adjusted the score accordingly.

---

> > > ### Author Response · Authors · 2024-08-14
> > > **Thanks for your valuable feedback!**
> > >
> > > Thank you for your reply and the recognition of our work. Your feedback is very important for us to further improve our paper. Thank you once again.

---

### Official Review · Reviewer_TdHL · 2024-07-07

**Soundness:** 3
**Presentation:** 3
**Contribution:** 3
**Rating:** 7
**Confidence:** 2

**Summary:**

This work is concerned with LLMs planning abilities in agent datasets. Instead of only fine-tuning the agent model on expert trajectories, they add “task knowledge” information. This information is created by comparing reject trajectories and expert trajectories, following previous work (NAT; Wang et al., 2024). In addition, the agent model is prompted to summarize the state, which helps avoid generating invalid actions.

**Strengths:**

* The idea of explicating the preference trajectories data (NAT; Wang et al., 2024) into task knowledge is interesting (subsection 3.1 and case study in Appendix F).
* Strong results when combining both task knowledge and state knowledge.
* Weak-guide-strong analysis (table 4) interestingly shows the benefits of explicating the trajectories preference knowledge.

**Weaknesses:**

- When removing the state knowledge (figure 3), it seems that this approach does not outperform NAT, which uses SFT on the same trajectories preference data.

- Missing qualitative analysis of the generated task knowledge.

**Questions:**

**Suggestions**
* Case study in Appendix F is very important to understand the paper, providing intuition and a qualitative analysis which explains why preference trajectories data is useful. I would put it in the main paper to guide the reader.

**Questions**
* Line 135 mentions that the WKM and the agent are the same backbone model. What about the LM that generates the reject trajectories, is it the same?

**Limitations:**

The authors discuss limitations of their work.

---

> ### Author Rebuttal · Authors · 2024-08-04
>
> We are deeply grateful for your valuable time and insightful feedback. Below are our detailed responses to your concerns.
>
> **Q1: When removing the state knowledge (figure 3), it seems that this approach does not outperform NAT, which uses SFT on the same trajectory preference data.**
>
> We greatly appreciate your meticulous observation. **However, in fact, even without the state knowledge, our method still outperforms NAT.** We feel so sorry that maybe the distance between Table 1 and Figure 3 in the paper has misled your judgment and we will improve this in our revision. Here we list the specific values in the table below for your convenience in observation:
>
> | Mistral-7B    | ALFWorld seen | ALFWorld unseen | WebShop | SciWorld seen | SciWorld unseen |
> | ------------- | ------------- | --------------- | ------- | ------------- | --------------- |
> | NAT           | 64.43         | 68.96           | 61.01   | 57.12         | 50.79           |
> | WKM w/o state | 69.29         | 75.37           | 63.68   | 60.81         | 53.42           |
> | WKM w/o task  | 67.86         | 70.67           | 62.44   | 55.04         | 51.52           |
> | WKM           | 73.57         | 76.87           | 65.48   | 62.12         | 53.62           |
>
> **Q2: Qualitive analysis and Appendix F**
>
> We feel so sorry that due to page limitations and the lengthy trajectories of the case in Figure 9, we did not include Appendix F in the main paper initially. **We will incorporate the textual analysis in Appendix F into the main paper in the upcoming revision.** In fact, on the right side of Figure 2, we also display part of a case's steps, from which you can clearly see the role and effectiveness of our task and state knowledge.
>
> **Q3: Line 135 mentions that the WKM and the agent are the same backbone model. What about the LM that generates the reject trajectories, is it the same?**
>
> Yes, you are right. They all share the same backbone model but with different LoRAs. All the training involved in our paper is based on LoRA, which allows the knowledge model and the agent model to be plug-and-play, significantly saving computational power while making the model switch and extend more flexibly and efficiently.
>
> Thank you again for your constructive suggestions!
>
> **Please let us know if you have any further questions, as we are happy to continue the discussion. If you find that our response addresses your concerns, would you kindly consider raising your rating score for our paper? We greatly appreciate your consideration.**

---

> > ### Comment · Reviewer_TdHL · 2024-08-10
> > **Response to rebuttal**
> >
> > I appreciate the authors' attention to my concerns and their efforts to clarify any misunderstandings.
> >
> > Concerning my initial issue, I believe the confusion arose because Figure 3 was labeled "w/ state" instead of "w/o task," which led me to misinterpret the results. The labels in the table provided by the authors offers a clearer perspective in my opinion, and I now recognize that their approach without state indeed outperforms NAT.
> >
> > I have adjusted my score to reflect my revised understanding.

---

> > > ### Author Response · Authors · 2024-08-12
> > > **Thank you for your in time feedback!**
> > >
> > > We are very delighted that our response can address your concerns, and we sincerely appreciate your recognition of our work!
> > >
> > > We have indeed realized that in the ablation study, the "w/o" may convey a clearer message than "w/", and we will make a corresponding revision to further enhance the readability of our paper.
> > >
> > > Thank you again for your prompt feedback!

---

### Official Review · Reviewer_pvgu · 2024-07-10

**Soundness:** 4
**Presentation:** 3
**Contribution:** 3
**Rating:** 6
**Confidence:** 4

**Summary:**

The paper presents a parametric World Knowledge Model (WKM) to enhance agent planning by providing both global prior task knowledge and local dynamic state knowledge.
Traditional LLMs often perform trial-and-error actions and generate hallucinatory actions due to their limited understanding of the physical world.
By imitating human cognitive processes, the WKM synthesizes knowledge from expert and sampled trajectories to guide agents more effectively.
Experimental results with state-of-the-art LLMs (Mistral-7B, Gemma-7B, and Llama-3-8B) show that this method improves performance on complex real-world tasks and reduces blind trial-and-error and hallucinatory actions.

**Strengths:**

WKM mimics human mental models, incorporating both global prior task knowledge and local dynamic state knowledge, and offers a novel and effective solution to the limitations of traditional LLMs in understanding the physical world. The human-guided action search would eliminate a lot of inappropriate hallucinations.

The method is rigorously tested on various complex real-world simulated datasets using Mistral-7B, Gemma-7B, and Llama-3-8B. The superior performance compared to strong baselines demonstrates the practical effectiveness and robustness of the proposed WKM, providing solid evidence for its advantages. Even these "small language models" prove the effectiveness of WKM.

This simulated knowledge base helps in better guiding, planning and assisting local planning, significantly improving the agent's overall performance and understanding of tasks.

**Weaknesses:**

The approach heavily relies on expert trajectories to synthesize both task and state knowledge. It takes a lot of effort to obtain such a database and pre-train/fine-tuning and off-line WKM.

WKM depends on the world dynamic as well as human demonstration.  In my opinion, the three "real-world simulated planning datasets":
ALFWorld, WebShop, and ScienceWorld are not all real. The ALFWorld and ScienceWorld are simulated environments with a discreet Embodied mechanism. The methodology of WKM may not be easily adapted to problem-solving in real scenarios.

**Questions:**

See weeknesses

**Limitations:**

no potential negative societal impact

---

> ### Author Rebuttal · Authors · 2024-08-04
>
> We are deeply grateful for your valuable time and insightful feedback. Below are our detailed responses to your concerns.
>
> **Q1: The approach heavily relies on expert trajectories to synthesize both task and state knowledge.**
>
> In fact, most mainstream agent planning methods currently either rely on proprietary models like GPT-4 (e.g., AutoGuide) or utilize open-source models for incremental training on expert trajectories and explored trajectories through techniques such as SFT or DPO (e.g., ETO, NAT). These approaches do not consume fewer resources (be it monetary or computational resources) than our method. **Our method, without reliance on GPT-4, enables small-scale open-source models to autonomously synthesize knowledge and train into parameterized knowledge models.** This small-scale knowledge model can not only guide fine-tuning-based agents of the same scale (Table 1) but also enhance powerful proprietary models like GPT-3.5/4 (Table 4). Moreover, the multi-task unified WKM has demonstrated superior performance (Figure 5), providing insights into the path toward exploring AGI in the future.
>
> **Q2: The methodology of WKM may not be easily adapted to problem-solving in real scenarios.**
>
> The current mainstream LLM-based agent planning benchmarks are all based on simulated environments of the real world. The three datasets we utilize—ALFWorld, WebShop, and SciWorld—are also commonly used in LLM-based agent planning. We understand the significance of applying the World Knowledge Model (WKM) to real-world scenarios, and we are actively working towards this, such as exploring a unified knowledge model that can be generalized to real-world situations. We hope to pair this with a generalizable unified agent model to achieve true AGI (Artificial General Intelligence). Although this path is still a long way off, our work, like the work of others, is striving towards this goal.
>
> Thank you again for your constructive suggestions!
>
> **Please let us know if you have any further questions, as we are happy to continue the discussion. If you find that our response addresses some of your concerns, would you kindly consider raising your rating score for our paper? We greatly appreciate your consideration.**

---

### Official Review · Reviewer_vJYv · 2024-07-28

**Soundness:** 2
**Presentation:** 2
**Contribution:** 2
**Rating:** 4
**Confidence:** 3

**Summary:**

This paper introduces a parametric World Knowledge Model (WKM) to enhance agent planning by integrating both global task knowledge and dynamic state knowledge. The authors claim that their approach can mitigate issues like blind trial-and-error and hallucinated actions in large language model (LLM) agents. They demonstrate the effectiveness of WKM through experiments on three real-world simulated datasets (ALFWorld, WebShop, and ScienceWorld) using state-of-the-art LLMs, showing superior performance compared to strong baselines.

**Strengths:**

1. The paper is well-written and easy to follow.
2. The motivation for the methodology is clear and logically presented.

**Weaknesses:**

My primary concerns with this paper are related to the methodology's implementation and the sufficiency of the experiments. Specific issues are detailed in the following questions.

**Questions:**

Methodology:

1. The motivation for rejecting trajectories is not sufficiently justified. (1) Why not and What if directly derive task knowledge from expert trajectories?  (2) Why assume that agent-generated trajectories are always inferior to the dataset trajectories? Could they not be better, and what would be the impact of this assumption? (3) Even if the above assumption holds, if agent-generated trajectories are always rejected, how can the authors claim, "Our purpose is to extract superior task knowledge that cannot be acquired solely through supervised fine-tuning on chosen trajectories, thus further effectively boosting the agent’s capabilities" (line 107-108)? (4) Ignoring the previous issues, in Line 104, the authors train an experienced agent to generate reject trajectories. However, trajectory generation requires an environment and an experienced agent. How can trajectories be generated without training the environment model? Is that done by interacting with the environment?

2. The definition of state knowledge is ambiguous. Lines 119-120 suggest that state knowledge is a local summarization of the policy function to instruct actions, but lines 123-124 define it as part of the MDP's state space. Besides, Figure 12's prompt also does not define state knowledge yet asks the LLM to generate it (How can an LLM generate knowledge without a definition?)

3. The knowledge model is trained to output task knowledge and state knowledge using data directly labeled by the LLM through prompts. Why not directly use these prompts to output task knowledge and state knowledge instead of retraining a model?

Experiments:

1. Can the paper "AutoGuide: Automated Generation and Selection of State-Aware Guidelines for Large Language Model Agents" be a baseline of this paper?
2. Can the authors compare results with gamma=1 and gamma=0? It is essential to understand which part of the output action has the most significant impact.

---

> ### Author Rebuttal · Authors · 2024-08-04
>
> We are deeply grateful for your valuable time and insightful feedback. Below are our detailed responses to your concerns.
>
> **Q1: The motivation for rejecting trajectories is not sufficiently justified.**
>
> In fact, regarding your concern about the rejected trajectories, **we have explained in lines 113-117 of the paper**. We will consolidate this with lines 102-108 to make it more convenient for readers. We sincerely apologize for any inconvenience this has caused you. Below is a further explanation addressing your concern:
>
> (1) To intuitively demonstrate the effect of introducing rejected trajectories, we conduct experiments synthesizing task knowledge based solely on expert trajectories. The advantage of introducing rejected trajectories is very significant:
>
> |Mistral-7B|ALFWorld seen|ALFWorld unseen|WebShop|SciWorld seen|SciWorld unseen|
> |-----------------|-------------|---------------|-------|-------------|---------------|
> |w/o rejected traj|67.19|71.57|63.97|55.49|48.38|
> |WKM|73.57|76.87|65.48|62.12|53.62|
>
> (2) The assumption is based on the premise that expert trajectories are manually labeled, and they always achieve the best in quality and final reward. As we mentioned in lines 113-117 since expert trajectories are gold, their final reward $r(u,\tau_w)$ always satisfies $r(u,\tau_w)=1$. Therefore, the final reward $r(u,\tau_l)$ of agent-generated trajectories always satisfies $r(u,\tau_l) \leq r(u,\tau_w)$. If $r(u,\tau_l) < r(u,\tau_w)$, there is no doubt that expert trajectories are better; if $r(u,\tau_l) = r(u,\tau_w)$, **since expert trajectories are gold with no excess planning steps**, they are shorter and more efficient. We want the agent model to learn how to perform more efficient planning and avoid blind trial-and-error from these trajectories, thus we also consider agent-generated trajectories to be rejected.
>
> (3) Our experienced agent has undergone SFT on the chosen (expert) trajectories. As we all know, training on a dataset doesn't mean that the model has fully learned all the knowledge in that dataset. Therefore, we let the experienced agent run through the training data again to generate explored trajectories, so that the agent can summarize knowledge that cannot be learned solely through SFT. This step is a little similar to DPO, but we achieve it through knowledge augmentation rather than directly converting it into a loss calculation between chosen and rejected trajectories.
>
> (4) Yes, the generation of rejected trajectories involves direct interaction between the experienced agent model and the environment of the training set.
>
> **Q2: The definition of state knowledge is ambiguous.**
>
> Our state knowledge here is a natural language description of the current environment and the agent's state, so we define it as a part of the MDP's state space. As shown in Fig 12, we teach the agent to summarize state knowledge through few-shot examples, rather than zero-shot instruction. In the main paper Fig 2 and Appx F Fig 9, we provide some cases where you can see the specific appearance of state knowledge.
>
> **Q3: Why not directly use prompts to output task and state knowledge instead of retraining a model?**
>
> Firstly, task and state knowledge are obtained with the need for expert trajectories. Since we cannot obtain expert trajectories on the test set, it's hard to provide high-quality knowledge through prompts. Secondly, even if we use the knowledge obtained from the training set as few-shot prompts, it is evident that training a model is more generalizable than simply using prompts. In fact, **as one of our baselines, the knowledge for KnowAgent is provided through prompts and our dataset-level knowledge analysis in Fig 4 also uses prompts to provide knowledge. WKM's performance is significantly better.**
>
> To completely address your concern, we also conduct experiments using high-quality task and state knowledge summarized from the training set as few-shot prompts. The experimental results can once again prove that training a model performs better:
>
> |Mistral-7B|ALFWorld seen|ALFWorld unseen|WebShop|SciWorld seen|SciWorld unseen|
> |----------------|-------------|---------------|-------|-------------|---------------|
> |prompt knowledge|65.14|67.40|61.03|56.36|45.27|
> |WKM|73.57|76.87|65.48|62.12|53.62|
>
> **Q4: AutoGuide as a baseline**
>
> In fact, we greatly wanted to include AutoGuide as a baseline when we began our experiments. **However, at that time and even now, it doesn't have open-source code available.** We attempted to replicate it solely based on its paper, but some implementation details including some specific prompts cannot be obtained solely from the paper. In fact, as a prompt-based baseline, it relies on the strong GPT-4. And using 7/8B models would significantly degrade its performance, making it far less effective than fine-tuning-based methods, not to say our WKM.
>
> **Q5: Compare results with $\gamma=1$ and $\gamma=0$**
>
> $\gamma=1$ is equivalent to removing state knowledge, and **our ablation experiment (Figure 3 w/ task) already includes this scenario**.
>
> We further conduct experiments specifically for $\gamma=0$, comparing it with $\gamma=1$ and WKM:
>
> |Mistral-7B|ALFWorld seen|ALFWorld unseen|WebShop|SciWorld seen|SciWorld unseen|
> |----------|-------------|---------------|-------|-------------|---------------|
> |$\gamma=0$|1.58|0.00|25.83|18.69|15.37|
> |$\gamma=1$|69.29|75.37|63.68|60.81|53.42|
> |WKM|73.57|76.87|65.48|62.12|53.62|
>
> It can be observed that state knowledge primarily serves as a constraint to alleviate hallucinated actions for the agent model. **However, when we fully trust it($\gamma=0$), its lack of generalization significantly harms the performance of the agent model.**
>
> Thank you again for your constructive suggestions!
>
> **Please let us know if you have any further questions. If you find that our response addresses some of your concerns, would you kindly consider raising your rating score for our paper? We greatly appreciate your consideration.**

---

> > ### Comment · Reviewer_vJYv · 2024-08-11
> >
> > Thank you for your response, which addressed some of my concerns.
> >
> > **Follow-up question for Q1:**
> > Thank you for the clarification. I have a follow-up question: Why does the method with task knowledge *only* perform better than the DPO-style baselines even in seen tasks? Since your expert trajectories are optimal, at least in the seen tasks, the fine-tuning method should be the better one as it only needs to memorize the given data.
> >
> > **Follow-up question for Q3:**
> > I would like to clarify my question. In Equation (5), state knowledge is constructed using LLM by prompting with $\rho_{stateKnow}$ and $h_t$. However, in Line 157, state knowledge is generated from $\pi_\phi(\cdot|h_t)$, i.e., the World Knowledge Model. Why not use the same method as in Equation (5) directly in Line 157 as they share the same inputs?
> >
> > **Additional Suggestions:**
> > After the author's clarification, I understand more about the methodology. However, I find the writing to be informal. Despite the trend towards more relaxed expressions, I believe that for a NeurIPS paper, the presentation should be more rigorous. For example,
> > 1. "Subsequently, the experienced agent explores the training set tasks again to generate rejected trajectories. Our purpose is to extract superior task knowledge that cannot be acquired solely through supervised fine-tuning on chosen trajectories, thus further effectively boosting the agent’s capabilities." As clarify by the authors, there is a significant logical gap between these two sentences. Generating rejected trajectories alone does not lead to extracting superior task knowledge that cannot be acquired through supervised fine-tuning on chosen trajectories.
> > 2. "The state knowledge serves as the dynamic knowledge to constrain the agent model’s local planning and prevent it from generating hallucinatory actions." If you are talking about MDP, it is incorrect to say the state constrains the agent’s ... abilities. The state is just defined to provide full information of the environment for the policy to make decisions [1], rather than enhancing or constraining the agent’s abilities.
> >
> > [1] Reinforcement Learning: An Introduction.

---

> ### Author Response · Authors · 2024-08-11
> **Thanks for your response!**
>
> We are very fortunate that our rebuttal could address some of your questions and are extremely grateful for the opportunity to engage in further discussion with you. In response to your follow-up questions, we would like to provide the following explanations:
>
> **Follow-up question for Q1**
>
> You may consider our approach to be somewhat similar to the idea behind DPO, but while DPO optimizes the model by comparing the losses between chosen and rejected trajectories, requiring the memorization of both types of data, our method contrasts chosen and rejected trajectories to summarize knowledge. During training, the model only needs to learn from this knowledge without the additional requirement to memorize the rejected trajectories. This also validates that our knowledge enhancement allows the agent model to acquire knowledge that cannot be learned through SFT loss calculations.
>
> **Follow-up question for Q3**
>
> We would like to address this question from two perspectives:
>
> 1. The notation $h_t$ in Equation (5) is defined according to Equation (1), which does not include the task knowledge $\kappa$. However, **as mentioned in line 152, we have redefined $h_t$ to include the task knowledge $\kappa$**. Therefore, the inputs defined in Equation (5) and line 157 are different. We apologize if the redefinition of $h_t$ was not clear and may have caused some confusion. We will emphasize this in the revision by bolding it to improve readability.
>
> 2. **Even if we disregard the changes of $h_t$, as we clarified in our rebuttal Q2, the prompt to summarize state knowledge ($\rho_{\rm StateKnow}$) in Equation (5) are essentially few-shot examples.** This aligns with the experimental setup we supplemented in the rebuttal for Q3 if we directly use $\rho_{\rm StateKnow}$ at the inference time, and the results of our experiments have demonstrated that the performance of providing knowledge through prompts is poor.
>
> **For Additional Suggestions**
>
> We sincerely apologize for any confusion caused in your reading.
>
> 1. By separating the synthesis of task knowledge into two processes, our initial intention was to clarify the logic, but we did not anticipate the difficulties it might cause for you. We are truly sorry for this oversight. As we have clarified, we will integrate lines 102-117 in the revision to achieve better readability and make it easier for readers to understand.
>
> 2. Initially, we defined the state knowledge within the state space to facilitate comprehension for readers. This definition might indeed lack rigor, and we will redefine it with another symbol similar to the task knowledge in the revision to ensure clearer expression. We are genuinely sorry for any inconvenience this has caused you!
>
> Once again, we greatly appreciate your feedback. Your comments are invaluable to the improvement of our work.
>
> **We look forward to your further reply and would like to continue the discussion. Would you kindly consider raising your rating score for our paper if you find that our response addresses some of your concerns? We greatly appreciate your consideration.**

---

> ### Comment · Reviewer_vJYv · 2024-08-11
>
> Thank you for the prompt response.
>
> Follow-up question for Q1: I would like to clarify my question again. This question follows up on your statement: "Therefore, we let the experienced agent run through the training data again to generate explored trajectories, so that the agent can summarize knowledge that cannot be learned solely through SFT. This step is somewhat similar to DPO, but we achieve it through knowledge augmentation rather than directly converting it into a loss calculation between chosen and rejected trajectories." It is somewhat intuitive, but following this line of reasoning, using DPO directly should yield better results compared to the WKM variant w/o state knowledge, i.e., WKM w/ task in your ablation studies, at least in the seen tasks. However, it seems contradictory that we observe the opposite result.
>
> Follow-up question for Q3: Could the authors clarify why Equation (5) (with your redefined ${h_t}$) cannot be used to output the state knowledge? Additionally, I do not grasp the idea of few-shot prompts, as using Equation (5) to output state knowledge does not necessarily depend on them. Generally speaking, why can we assert that using model ${A(y|X)}$ to label a dataset ${D=\{(X,y)}\}$, and then fine-tuning ${A}$ with ${D}$ to obtain ${A'}$, results in a model ${A'}$ that is superior to ${A}$?

---

> ### Author Response · Authors · 2024-08-11
> **Thank you for your continuous feedback!**
>
> Here is our further clarification of your questions:
>
> 1. We sincerely apologize for any confusion and we need to sincerely say that we still have not understood why you suggested that our statement could reason that DPO should perform better on seen tasks. We need to reiterate that our method enables the model to read both correct and incorrect trajectories for the same task, using $\rho_{\rm TaskKnow}$ (detailed in Figure 11) to summarize the knowledge why the correct trajectories are better than the incorrect ones, and training the WKM to learn to generate this kind of knowledge to augment the agent model. On the other hand, DPO directly trains the agent model by minimizing the loss (increasing the distribution gap between correct and incorrect trajectories) to favor the generation of correct trajectories. Therefore, we believe there is no direct relationship or conflict between the two approaches. Our previous statement of "similar to DPO" might have caused misunderstanding; it only proves that the improvement of our agent model augmented by synthetic task knowledge is better than training the agent model with DPO.
>
> 2. **(1)** As mentioned in lines 122-123, the prompt $\rho_{\rm StateKnow}$ that we use to summarize state knowledge is detailed in Appendix I.2, which is displayed in Figure 12. In response to Q2, we have clarified to you that the key part of our prompt is the State Knowledge Example (colored in purple), which is actually the few-shot examples. Therefore, we stated that generating state knowledge directly with Equation (5) is equivalent to the experiment we supplemented in Q3, where knowledge was generated directly using few-shot examples. As shown in our previous table, while it is certainly possible to generate knowledge by few-shot examples, it is clear that its effectiveness is not as good as that of the WKM.
>
>
>    **(2)** Regarding the question you raised about why training Model A with data annotated by Model A itself can lead to improvements for A, **this issue has been extensively validated in the field of LLM synthetic data (also known as self-training) [1][2][3][4][5][6]**. The key to self-training lies in ensuring the quality of self-generated data. For instance, [3] achieves this by implying the correct labels of the data to the model, while [6] relies on the model's self-judgment capability to assess the quality of the data. Our approach ensures the quality of synthesized knowledge by meticulously designing few-shot examples and using entirely correct expert trajectories. The advantage of this method is that it does not depend on a large amount of manually annotated data or powerful closed-source models (e.g., GPT-4), fully leveraging the model's own potential. However, this method usually has an upper-bound because we cannot guarantee that the synthesized data is 100% correct. As the model generates more data, the diversity of the data is also likely to decrease. This is a key research direction in the field of data synthesis today, that is, how to improve the diversity and quality of synthesized data. The principle behind self-training is still under investigation; we speculate that it may be because the data annotated by the model itself is more in line with the "data distribution" understood by the model. From the model's perspective, this distribution may be smoother. Although this method may not be as effective as training on fully human-annotated data, the most important aspect is that it solves the resource consumption brought about by a large amount of manual data annotation. And if one day human-annotated data is exhausted or LLM grows into a super model that humans cannot supervise, we can only rely on the data synthesized by the model itself to improve the ability of LLM, so this is a very promising research direction in the LLM community.
>
>    [1] On LLMs-Driven Synthetic Data Generation, Curation, and Evaluation: A Survey.
>
>    [2] KnowAgent: Knowledge-Augmented Planning for LLM-Based Agents.
>
>    [3] STaR: Self-Taught Reasoner Bootstrapping Reasoning With Reasoning.
>
>    [4] Best Practices and Lessons Learned on Synthetic Data for Language Models. (Google DeepMind)
>
>    [5] Self-training Language Models for Arithmetic Reasoning.
>
>    [6] Self-Rewarding Language Models.
>
> We look forward to your further reply and would like to continue the discussion.
>
> Thanks again!

---

> > ### Comment · Reviewer_vJYv · 2024-08-12
> >
> > As the rebuttal period is concluding, I would like to finally point out that self-training should not be interpreted in the same manner as the authors propose. Aligning with the notations in our discussion, for [3], in order to label $y$ in $D=\\{ (X,y) \\}$ for fine-tuning, (1) they filter the generated rationales ($y$ includes rationales followed by an answer in this setting) from model $A$ to retain only those that result in correct answers; (2) for incorrect answers, they provide the correct answer as a hint to the model and ask model $A$ to generate rationales in the same style as in the previous rationale generation step, instead of using the label $y$ generated by model $A$ directly. For [6], they employ RL optimization, where rewards are assigned by model $A$ based on the label $y$ generated by model $A$. This approach is still intuitive, as studies have shown that a model's evaluation capability generally surpasses its generation capability [7]. Similarly, they do not use the label $y$ generated by model $A$ directly for fine-tuning.
> >
> > I would like to thank the authors' detailed response and the efforts made in the rebuttal. At this point, I maintain my original concerns; however, I am open to reconsidering my evaluation based on the suggestions from the Area Chairs.
> >
> > [7] Language Model Self-improvement by Reinforcement Learning Contemplation

---

> ### Author Response · Authors · 2024-08-12
> **Thank you for your patient response!**
>
> Dear reviewer,
>
> We greatly respect your position, but we have to say that we still insist on our viewpoint. In the era of deep learning, we cannot say that a method cannot be used because it may lack some interpretability despite being effective. Moreover, **our goal is to train a World Knowledge Model that provides both task knowledge and state knowledge**. High-quality task knowledge requires comparison between positive and negative examples to be obtained, which cannot be provided solely through prompts (as we have demonstrated in our supplementary experiments 1 and 2, our comparison with KnowAgent, and our analysis in Figure 4). Therefore, **it would be contrary to our approach of training a unified WKM if we were to train a model to provide task knowledge while prompting another model to provide state knowledge**.
>
> With still two days remaining until the end of the discussion period, we also warmly welcome any further questions you may have. At the same time, we are very grateful for your patient review of our responses during the rebuttal phase. Thank you once again!

---

### Author Rebuttal · Authors · 2024-08-04

Dear all reviewers,

Thank you for your thoughtful reviews! We appreciate all of your **positive comments** highlighting the strengths of our work for a summary:

## **Our Strengths Summarized by Reviewers**

- **Reasonable motivation**:
  - "The motivation for the methodology is clear and logically presented."(reviewer vJYv)
  - " It's also quite reasonable to introduce a world knowledge model to enhance task-specific knowledge during planning."(reviewer 8XFL)
- **Interesting, novel, and easy to follow**:
  - "The paper is well-written and easy to follow."(reviewer vJYv)
  - "offers a novel and effective solution"(reviewer pvgu)
  - "The idea of ... is interesting."(reviewer TdHL)
  - "the method is novel"(reviewer 8XFL)
- **Superior performance, promising results, and interesting analysis**:
  - "The superior performance compared to strong baselines demonstrates the practical effectiveness and robustness of the proposed WKM, providing solid evidence for its advantages."(reviewer pvgu)
  - "Strong results when combining both task knowledge and state knowledge."(reviewer TdHL)
  - "Weak-guide-strong analysis (table 4) interestingly shows the benefits of explicating the trajectories preference knowledge."(reviewer TdHL)
  - "It also shows promising results using only 7B models that could barely perform planning with basic methods."(reviewer 8XFL)
- **Well-written** (reviewer vJYv, reviewer 8XFL)

## **Our Supplement Experiments**

We also sincerely thank reviewers for your constructive feedback and questions to improve our manuscript. In response to the reviewers' questions, **we mainly supplement the following experiments**:

1. For reviewer vJYv: **To intuitively demonstrate the effect of introducing rejected trajectories**, we conducted experiments synthesizing task knowledge based solely on expert trajectories.
2. For reviewer vJYv: **To address the concerns about the necessity to retrain a knowledge model**, we conduct experiments using high-quality task and state knowledge summarized from the training set as few-shot prompts for knowledge models to provide knowledge.
3. For reviewer vJYv: **For the convenience of understanding which part of the output action has the most significant impact**, we compare the results with $\gamma=0$, $\gamma=1$, and our WKM.
4. For reviewer 8XFL: **To fully show the advantages of separate training**, we further conduct experiments where a single model learns planning and knowledge at the same time.

**We have added supplementary experiments 1, 2, and 4 as parts of our Ablation Study and revised our Figure 3. We have also added supplementary experiment 3 to our Appendix with a table. The revised Figure 3 and the added $\gamma$ analysis table can be seen in our submitted PDF.** We will add the corresponding textual analysis in our main paper.

We will continue to further enhance the quality of this paper according to the discussion with reviewers.

Last but not least, we wish to **reiterate the motivation and main contributions** of our paper.

## **Our Motivation**

As most state-of-the-art LLMs are autoregressive models trained with next-token prediction, they lack the ability to essentially understand the real world, leading to generating hallucinatory actions in local planning and performing brainless trial-and-error in global planning. In contrast to LLMs, humans possess a mental knowledge model about the physical world. When facing a specific task, they will first briefly rehearse the entire process in mind using their rich prior knowledge and constantly maintain a dynamic cognition of the current world state. **The process by which humans handle planning tasks reminds us to develop a parametric World Knowledge Model (WKM) to facilitate agent planning.**

## **Our Contributions**

- Imitating humans' mental knowledge model, we introduce parametric World Knowledge Model (WKM), providing prior task knowledge to guide the global planning and dynamic state knowledge to assist the local planning.
- Experimental results on three complex real-world simulated datasets with three state-of-the-art open-source LLMs, Mistral-7B, Gemma-7B, and Llama-3-8B, demonstrate that our method can achieve superior performance compared to various strong baselines.
- We analyze to illustrate that our WKM can effectively alleviate the blind trial-and-error and hallucinatory action issues, providing strong support for the agent’s understanding of the world.
- Other interesting findings of our paper include: 1) our instance-level task knowledge can generalize better to unseen tasks, 2) weak WKM can guide strong agent model planning, and 3) unified WKM training has promising potential for further development.

## **Thank you!**

We sincerely thank the reviewers for their constructive suggestions and questions to enhance our paper. Please reply if you have any further questions, and we will be more than happy to continue the discussion.

---

### Decision · Program_Chairs · 2024-09-25

**Decision:**

Accept (poster)

**Comment:**

The authors provide task knowledge and state knowledge to improve global and local planning in smaller LLMs and shows promising generalization. The paper is well written with many experiments and additional analyses that strengthens their novel approach. The authors and reviewers have engaged in thorough discussions, and in spite of some disagreement between the authors and one of the reviewers, the paper is recommended for acceptance as a poster, conditional on integrating the new results and responses to reviewers in the paper.